# The role of TyG index as a predictor of all-cause mortality in hospitalized patients with acute pancreatitis: a retrospective study utilizing the MIMIC-IV database

**Jian Liao**[1☯], **Dingyu Lu**[2☯], **Hong Xie**[1], and **Maojuan Wang**[1]*

**1** Intensive care Unit, Deyang People's Hospital, Deyang, China, **2** Oncology Department, Deyang People's Hospital, Deyang, China

☯ These authors contributed equally to this work.

* wangmj05@sina.com

## Abstract

### Background

The TyG index is widely recognized as a reliable indicator for cardiovascular disease risk and as a biomarker for assessing insulin resistance(IR). However, its significance in the context of patients with acute pancreatitis(AP) needs further exploration. This study aimed to investigate the association between the TyG index and the risk of all-cause mortality in critically ill patients with AP.

### Methods

Data for this retrospective study were obtained from the MIMIC IV2.2 database. The participants were divided into four groups based on the TyG index tertiles. The primary outcome measured was in-hospital all-cause mortality. We employed Cox proportional hazards regression analysis and restricted cubic splines to evaluate the correlation between the TyG index and clinical outcomes in patients with AP.

### Results

The study included 586 patients, of which 44.71% were male. The rates of mortality observed in the hospital stay and in the ICU stay were 19.28% and 12.97%. By conducting multivariable Cox proportional hazards, it was determined that the TyG index was independently associated with a heightened risk of in-hospital mortality [HR(95%CI) of 1.38(1.03-1.87, P=0.033)] and in ICU mortality [1.65(1.12-2.44), P=0.012]. The analysis using restricted cubic splines showed that there was a consistent and gradually increasing risk of all-cause mortality as the TyG index increased. This indicates that a higher TyG index is associated with a higher risk of mortality.

### Conclusion

In critically ill patients with AP, the TyG index shows a notable correlation with all-cause death in both hospital and ICU. The TyG index can be useful in identifying insulin

**Data availability statement:** The data underlying the results presented in the study are available from the MIMIC-IV repository. https://physionet.org/content/mimiciv/0.4/.

**Funding:** The author(s) received no specific funding for this work.

**Competing interests:** The authors have declared that no competing interests exist.

resistance at an early stage in patients with AP, thereby improving risk assessment and guiding subsequent interventions.

## Introduction

Acute pancreatitis (AP) is a disease that results in the activation of pancreatic enzymes, leading to the damage and inflammation of pancreatic tissue [1]. The disease progresses rapidly and presents with a variety of clinical manifestations, making early diagnosis challenging. The severity of systemic involvement can vary, ranging from mild to severe. Mild cases often exhibit edema and have a good prognosis. However, a small percentage of patients develop severe acute pancreatitis, which is characterized by hemorrhagic atrophy of the pancreas and is usually associated with complications such as infection, peritonitis, or shock. The pathogenesis of this disease is complex and effective treatment options are currently lacking. Although the cure rate has improved over time, the overall fatality rate remains high. [2]

Insulin resistance (IR) is a vital aspect in the onset and progression of acute pancreatic condition [3]. IR, a pathological physiological state, is distinguished by decreased insulin sensitivity in peripheral tissues. This condition is frequently spotted in sepsis patients, manifesting in elevated insulin levels and reduced sensitivity [4]. The connections between AP and metabolic disorders, including hyperlipidemia and hyperglycemia, have been expounded in various studies [5,6]. Surrogate markers for insulin resistance, such as the triglyceride-glucose (TyG) index, have gained attention. Research indicates that the TyG index is connected to the advancement of metabolic disorders [7]. Triglyceride TyG index has been identified as a significant risk factor for various health conditions, including cardiovascular disease (CAD), cerebrovascular disease (CVD), sepsis, and acute kidney injury in critically ill patients with heart failure [8–11]. These published studies have provided evidence that supports the association between triglyceride TyG index and the development of these diseases in this specific patient population. The findings suggest that monitoring and managing triglyceride TyG index could potentially help mitigate the risk of these adverse health outcomes. Further research in this area is warranted to better understand the underlying mechanisms and develop targeted interventions for improved patient outcomes.

A number of studies have demonstrated that an increased TyG index is strongly linked to a higher risk of all-cause mortality [12,13]. However, there is a scarcity of research investigating the relationship between TyG index values and the severity levels of acute pancreatitis (AP). In order to address this gap, a retrospective cohort study was conducted to investigate the potential of the TyG index in predicting all-cause mortality in critically ill AP patients.

## Methods

### Study population

The researchers conducted a retrospective observational study using data from the publicly accessible Medical Information Mart for Intensive Care IV (MIMIC-IV) database, which can be found at https://mimic.mit.edu. Specifically, the study examined the medical records of patients in the ICU at Beth Israel Deaconess Medical Center from 2008 to 2019 [14]. In this particular study, data were analyzed retrospectively using an observational design. Ding yu Lu, as one of the authors, fulfilled the prerequisites to gain access to the database and undertook the task of data extraction. The patient cohort for this research comprised individuals with a confirmed diagnosis of acute pancreatitis, following the guidelines outlined in the International Classification of Diseases, 9th and 10th Revision. Ethical review and approval were

waived for this study, due to REASON: The use of the MIMIC-IV database was approved by the review committee of Massachusetts Institute of Technology and Beth Israel Deaconess Medical Center. The data is publicly available (in the MIMIC-IV database), therefore, the ethical approval statement and the requirement for informed consent were waived for this study.

The research excluded individuals below the age of 18 during their initial admission, individuals who experienced multiple admissions to the intensive care unit (ICU) due to acute pancreatitis (only data from te first admission were considered); individuals with severe ailments like end-stage renal dysfunction, cirrhosis, or cancer, individuals with an ICU stay of less than 24 hours, and individuals who lacked adequate data (TG and glucose level) on their first day of admission were excluded. A total of 586 patients formed the final study cohort and were divided into four groups based on the quartiles of the TyG index observed on their first day in the ICU (Fig 1 ).

## Data collection

To conduct the data extraction, we utilized PostgresSQL (version 13.7.2) software and Navicate Premium (version 16) tool by employing Structured Query Language (SQL). The extraction process prioritized four distinct categories of potential variables: (1) demographic factors encompassing age, gender, weight, and BMI; (2)comorbidities such as heart failure, AKI_48h, respiratory failure, sepsis, diabetes, and hypertension; (3) laboratory parameters including white blood cells (WBC), hemoglobin, platelet count, albumin, triglyceride, glucose, sodium, potassium, lactate, fibrinogen, anion gap, and creatinine; and (4) severity of illness scores at admission, specifically the Acute Physiology Score III (APSIII), the simplified Acute Physiology Score II (SAPS-II), Systemic Inflammatory Response Syndrome (SIRS), and the

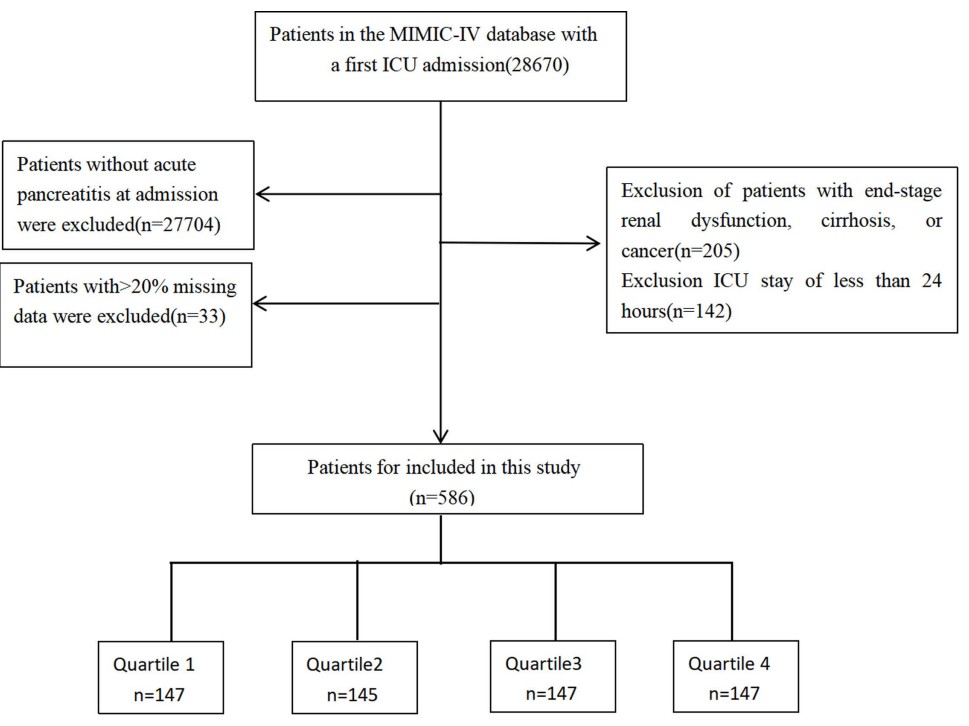

**Fig 1.** Flow of included patients through the trial.

Sepsis-related Organ Failure Assessment score (SOFA) [15,16]. The TyG index was calculated using the following formula: ln [fasting TG (mg/dl) ×fasting glucose (mg/dl)/2]. [12,17]

Exclusively acquired laboratory variables were obtained solely during the initial 24-hour period following patient admission. In instances where there were multiple outcomes, the average measurement was employed. To mitigate any potential bias, variables containing missing values surpassing 20% were eliminated. To handle variables with less than 20% missing data, the research team employed the multiple imputation (missForest R) technique (Additional file 1, S1 Table) [18,19].

## Outcomes

The main outcome of this study was hospital all-cause mortality. Secondary outcomes focused on mortality in the ICU.

## Statistical analysis

Continuous variables were presented as the mean±SD or median and interquartile range (IQR). The comparison of continuous variables was performed using t-test or ANOVA, or using Mann–Whitney U-test or Kruskal–Wallis test, as appropriate. Categorical variables were expressed as numbers or percentages (%), and their analysis was implemented by means of Fisher's exact test or Pearson chi-square test.

Kaplan-Meier survival analysis was used to assess the incidence rate of primary outcome events in different stratified groups based on the TyG index. The log-rank test was employed to examine any observed disparities. Binary logistic regression analysis was conducted to evaluate factors influencing the risk of all-cause death.

Univariable Cox regression analysis was conducted to investigate the association between the TyG index and the endpoints. Variables that showed clinical significance and had a significance level of P<0.05 were included in the multivariable Cox proportional hazards model. (Model 1: unadjusted; Model 2: adjusted for age, gender, and BMI; Model 3: adjusted for Age, BMI, Gender, WBC, PLT, HGB, FBG, Lac, Cre, K, TG, SOFA, AKI_48h, Hypertension, Heart failure, sepsis, Diabete, and Respiratory failure). HRs were counted and the findings were presented with 95% confidence intervals (CI). The lowest quartile of the TyG index was used as the baseline group in all four models.

We also conducted an analysis to examine the non-linear relationship between the baseline TyG index and ICU and in-hospital all-cause mortality. This analysis was done using a restricted cubic spline regression model with five knots.

Finally, subgroup analyses were conducted to investigate the consistency of the prognostic value of the TyG index across different subgroups. These subgroups were categorized based on age (≤65 versus >65 years), gender (female versus male), BMI (≤30 versus >30 kg/m²), and the presence of specific medical histories including Diabetes, AKI_48h, Sepsis, and Respiratory failure. Likelihood ratio tests were employed to evaluate the association between the TyG index and the variables used for stratification. The data analyses were conducted using R software (version 4.2.2) and SPSS statistical software (Version 25.0). For all analyses, a 2-side P<0.05 was considered statistically significant.

## Results

The study enrolled a total of 586 critically ill patients with acute pancreatitis from the MIMIC-IV database, as depicted in Fig 1.

Table 1 presents the baseline characteristics of all patients. The median age of the study participants was 56 years (interquartile range: 45-70 years), with males comprising 44.71% of

**Table 1. Baseline characteristics and outcomes of participants stratified by quartiles of TyG index[a].**

| Variable | Total (n = 586) | Q1 (n = 147) | Q2 (n = 145) | Q3 (n = 147) | Q4 (n = 147) | F/Z/X2 | P |
|---|---|---|---|---|---|---|---|
| Demographic | | | | | | | |
| Age, years | 56 (45 - 70) | 56(41 - 69) | 54 (44 - 67) | 58 (48 - 74) | 55 (46 - 68) | 7.535 | 0.057 |
| Gender, male, n(%) | 262 (44.7) | 68 (46.2) | 58 (40.0) | 72 (48.9) | 64 (43.5) | 2.61 | 0.456 |
| BMI, kg/m2 | 31.0 ± 5.7 | 30.0± 6.6 | 30.0 ± 5.4 | 31.0 ± 5.9 | 33.1 ± 4.2 | 9.56 | <.001 |
| Laboratory tests | | | | | | | |
| WBC, K/uL | 14.2 ± 8.5 | 15.7 ± 10.0 | 13.0 ± 7.7 | 13.8 ± 8.7 | 14.3 ± 7.2 | 2.56 | 0.054 |
| Platelet, K/uL | 217.3 ± 130.7 | 192.7 ± 109.2 | 211.9 ± 114.0 | 231.0±145.5 | 233.2±146.1 | 3.09 | 0.026 |
| Hemoglobin, g/dL | 11.5 ± 5.2 | 11.0 ± 2.3 | 11.5 ± 2.4 | 12.1 ± 9.7 | 11.5 ± 2.4 | 1.03 | 0.378 |
| Albumin, g/dL | 2.8 ± 0.5 | 2.8 ± 0.5 | 2.9 ± 0.5 | 2.8 ± 0.5 | 2.9 ± 0.5 | 1.14 | 0.331 |
| Sodium, mEq/L | 138.0 ± 5.8 | 138.4 ± 5.8 | 137.0 ± 5.7 | 138.4 ± 5.6 | 138.1 ± 6.1 | 1.82 | 0.141 |
| Potassium, mEq/L | 4.1 ± 0.8 | 4.1 ± 0.7 | 3.9 ± 0.6 | 4.0 ± 0.8 | 4.2 ± 1.0 | 3.61 | 0.013 |
| FBG, mg/dl | 156.23 ± 117.0 | 123.3 ± 45.3 | 129.6 ± 54.1 | 141.4 ± 73.3 | 230.2±192.3 | 30.83 | <.001 |
| Lactate, mg/dl | 1.9 (1.3 - 3.1) | 1.8 (1.2 - 3.0) | 1.7 (1.2 - 2.6) | 2.0 (1.3 - 3.7) | 2.0 (1.4 -3.4) | 9.40 | 0.024 |
| Fibrinogen, mg/L | 424.9 (309.2-518.9) | 439.9 (321.2-507.4) | 427 (287- 532) | 413 (30- 492) | 425 (312-530) | 1.21 | 0.749 |
| Anion gap, mEq/L | 16.14 ± 5.48 | 15.86 ± 4.67 | 15.70 ± 5.50 | 15.66 ± 5.61 | 17.32 ± 5.94 | 3.122 | 0.026 |
| Triglyceride, mg/dl | 438.5 ± 728.1 | 103.4 ± 45.4 | 265.2 ± 111.5 | 473.7± 168.1 | 909.3±1309.3 | 40.59 | <.001 |
| Creatinine, mg/dL | 1.52 ± 1.81 | 1.41 ± 1.55 | 1.69 ± 2.53 | 1.53 ± 1.63 | 1.46 ± 1.32 | 0.65 | 0.578 |
| Amylase, IU/L | 311.6 ± 301.6 | 313.5 ± 367.3 | 340.2 ± 293.0 | 282.6± 207.5 | 310.3± 316.6 | 0.89 | 0.446 |
| Commorbidities | | | | | | | |
| Heart failure, n (%) | 74 (12.63) | 18 (12.24) | 16 (11.03) | 21 (14.29) | 19 (12.93) | 0.73 | 0.866 |
| AKI_48hr, n (%) | 339 (57.85) | 92 (62.59) | 87 (60.00) | 76 (51.70) | 84 (57.14) | 3.93 | 0.268 |
| Respiratory failure, n (%) | 229 (39.08) | 59 (40.14) | 49 (33.79) | 60 (40.82) | 61 (41.50) | 2.31 | 0.509 |
| Sepsis, n (%) | 161 (27.47) | 41 (27.89) | 35 (24.14) | 47 (31.97) | 38 (25.85) | 2.51 | 0.473 |
| Diabete, n (%) | 160 (27.3) | 28 (19.05) | 34 (23.45) | 36 (24.49) | 62 (42.18) | 23.1 | <.001 |
| Hypertension, n(%) | 324 (55.29) | 77 (52.38) | 80 (55.17) | 73 (49.66) | 94 (63.95) | 6.84 | 0.077 |
| SOFA | 5(2- 8) | 4(2 - 6) | 5(2-8) | 7 (3 - 11) | 9 (4 - 14) | 2.27 | 0.032 |
| APSIII | 47 (33 - 65) | 46 (31- 69) | 43(32 - 64) | 48(37 -64) | 47.(35- 65) | 2.22 | 0.528 |
| SIRS | 3 (2- 4) | 3(2 - 4) | 3(2- 4) | 3 (3 - 4) | 3(3 - 4) | 0.86 | 0.834 |
| SAPSII | 33 (24 - 45) | 32 (23 - 44) | 31(22 - 47) | 35(27- 45) | 33 (25 - 46) | 3.62 | 0.304 |
| MV, n (%) | 237 (40.44) | 63 (42.86) | 52 (35.86) | 60 (40.82) | 62 (42.18) | 1.81 | 0.613 |
| CRRT, n (%) | 61 (10.41) | 19 (12.93) | 14 (9.66) | 11 (7.48) | 17 (11.56) | 2.64 | 0.449 |
| Outcomes | | | | | | | |
| Hospital_Los, Day | 10.4 (6.6-18.0) | 8.5(6.0-15.4) | 10.4 (6.7-18.0) | 11 (5.9-16.8) | 12.6 (7.7-21.3) | 12.96 | 0.005 |
| ICU_Los, Day | 3.3 (1.8- 6.8) | 3.0 (1.7- 6.3) | 3.1 (1. - 5.8) | 3.2 (2.1- 6.7) | 4.2 (2.0 - 8.9) | 7.85 | 0.049 |
| Hospital mortality, n(%) | 113 (19.28) | 18 (12.24) | 21 (14.48) | 27 (18.37) | 47 (31.97) | 22.11 | <.001 |
| ICU mortality, n (%) | 76 (12.97) | 11 (7.48) | 12 (8.28) | 16 (10.88) | 37 (25.17) | 26.70 | <.001 |

Data: N (%) or Mean (Q1–Q3) or mean±standard deviation.

[a]TyG index: Q1(8.56-9.20), Q2(9.20-9.95), Q3(9.95-10.57), Q4(10.57-11.09).

Abbreviation: BMI: Body Mass Index; WBC: White Blood Cell; fbg: fasting blood glucose; AKI_48hr: Acute Kidney Injury within 48 h; SOFA: Sequential Organ Failure Assessment; APSIII, Acute Physiology Score III; SIRS: Systemic Inflammatory Response Syndrome; SAPSII: Simplifed Acute Physiological Score II; Hospital_Los, Day: Hospital Length of Stay; ICU_Los, Day: Intensive Care Unit Length of Stay.

the sample. The hospital mortality rate was 19.28%, while the ICU mortality rate was 12.97%. The subjects were categorized into four groups based on their hospital admission TyG index levels: Quartile 1 (8.56-9.20), Quartile 2 (9.20-9.95), Quartile 3 (9.95-10.57), and Quartile 4 (10.57-11.09). Patients in the quartile 4 group exhibited higher body mass index and elevated

levels of Platelet, Na, Glucose, Lactate, Triglyceride. Furthermore, there was a higher prevalence of Diabetes and Hypertension among patients with a higher TyG index. Moreover, patients with a higher TyG index exhibited longer stays in the ICU(3.04 days vs. 3.1 days vs. 3.2 days vs. 4.2 days, P=0.049), and longer hospital stays (8.59 days vs. 10.4 days vs. 11.07 days vs. 12.66 days, P=0.005). The increase in TyG index was also associated with increased ICU mortality (25.17% vs. 10.88% vs. 8.28% vs. 7.48%, P<0.001) and hospital mortality (31.97% vs. 18.37% vs. 14.48% vs. 12.24%, P<0.001). Given the relatively strong association between the Q4 group and all-cause mortality, further comparisons were made between the Q4 and Q1-3 groups, and different grouping periods exhibited similar results (Additional File 2, Table S2).

Table 2 presents the baseline characteristics comparing survivors to non-survivors. The non-survivor group consisted of older individuals with more severe illness scores, a higher

**Table 2. Baseline characteristics of the Survivors and Non-survivors groups.**

| Variable | Total (n = 586) | Survivors (n=397) | Non-survivors (n=189) | t/z/X² | P |
|---|---|---|---|---|---|
| Demographic | | | | | |
| Age, years | 57.3±17.1 | 55.5±16.9 | 61.0± 16.8 | t=-3.683 | <.001 |
| Gender, male, n(%) | 262 (44.71) | 180 (45.34) | 82 (43.39) | $\chi^2$=0.198 | 0.657 |
| BMI, kg/m2 | 31.06 ± 5.77 | 30.72 ± 5.89 | 31.79 ± 5.45 | t=-2.105 | 0.036 |
| Laboratory tests | | | | | |
| WBC, K/uL | 14.26 ± 8.57 | 14.32 ± 8.95 | 14.13 ± 7.73 | t=0.262 | 0.794 |
| Platelet, K/uL | 217.3±130.7 | 220.8±132.24 | 209.9 ±127.4 | t=0.938 | 0.349 |
| Hemoglobin, g/dL | 11.57 ± 5.29 | 11.65 ± 6.20 | 11.39 ± 2.44 | t=0.564 | 0.573 |
| Albumin, g/dL | 2.89 ± 0.56 | 2.88 ± 0.54 | 2.91 ± 0.60 | t=-0.665 | 0.507 |
| Sodium, mEq/L | 138.0± 5.8 | 138.0± 5.9 | 137.9± 5.6 | t=0.314 | 0.754 |
| Potassium, mEq/L | 4.10 ± 0.84 | 4.06 ± 0.76 | 4.20 ± 0.98 | t=-1.833 | 0.068 |
| FBG, mg/dl | 156.2± 117.0 | 143.1± 100.9 | 183.6± 141.5 | t=-3.525 | <.001 |
| Lactate, mg/dl | 1.9 (1.30- 3.1) | 1.8 (1.2- 2.9) | 2.0 (1.4- 3.4) | Z=-2.056 | 0.040 |
| Fibrinogen, mg/L | 424.9 (309.2-518.9) | 425 (312.4-517.9) | 424.0 (302- 528.4) | Z=-0.253 | 0.800 |
| Anion gap, mEq/L | 16.14 ± 5.48 | 15.47 ± 4.97 | 17.54 ± 6.20 | t=-4.359 | <.001 |
| Triglyceride, mg/dl | 438.5±728.1 | 388.9± 664.0 | 542.8± 839.49 | t=-2.402 | 0.017 |
| Creatinine, mg/dL | 1.52 ± 1.81 | 1.42 ± 1.80 | 1.73 ± 1.81 | t=-1.957 | 0.051 |
| Amylase, IU/L | 311.6±301.6 | 313.3±309.9 | 307.9± 284.2 | t=0.204 | 0.838 |
| TyG Index | 9.8± 1.04 | 9.71 ± 0.95 | 10.28 ± 1.10 | t=-6.437 | <.001 |
| Commorbidities | | | | | |
| Heart failure, n (%) | 74 (12.63) | 52 (13.10) | 22 (11.64) | $\chi^2$=0.247 | 0.619 |
| AKI_48hr, n (%) | 339 (57.85) | 248 (62.47) | 91 (48.15) | $\chi^2$=10.769 | 0.001 |
| Respiratory failure, n(%) | 229 (39.08) | 148 (37.28) | 81 (42.86) | $\chi^2$=1.673 | 0.196 |
| Sepsis, n (%) | 161 (27.47) | 97 (24.43) | 64 (33.86) | $\chi^2$=5.713 | 0.017 |
| Diabete, n (%) | 160 (27.3) | 104 (26.20) | 56 (29.63) | $\chi^2$=0.760 | 0.383 |
| Hypertension, n(%) | 324 (55.29) | 213 (53.65) | 111 (58.73) | $\chi^2$=1.336 | 0.248 |
| SOFA | 5(2 - 8) | 5(2 - 8) | 6 (3 - 9) | Z=-2.315 | 0.021 |
| APSIII | 47 (33- 65) | 44 (31 - 59) | 53 (37 - 75) | Z=-4.136 | <.001 |
| SIRS | 3 (2 - 4) | 3 (2 - 4) | 3 (3- 4) | Z=-1.695 | 0.090 |
| SAPSII | 33 (24 - 45) | 31 (23 - 43) | 38 (27 - 51) | Z=-4.320 | <.001 |
| MV, n (%) | 237 (40.44) | 158 (39.80) | 79 (41.80) | $\chi^2$=0.213 | 0.645 |
| CRRT, n (%) | 61 (10.41) | 34 (8.56) | 27 (14.29) | $\chi^2$=4.494 | 0.034 |

Abbreviation: BMI: Body Mass Index; WBC: White Blood Cell; FBG: fasting blood glucose; AKI_48hr: Acute Kidney Injury within 48 h; SOFA: Sequential Organ Failure Assessment; APSIII, Acute Physiology Score III; SIRS: Systemic Inflammatory Response Syndrome; SAPSII: Simplifed Acute Physiological Score II; Hospital_Los, Day: Hospital Length of Stay; ICU_Los, Day: Intensive Care Unit Length of Stay.

incidence of sepsis and AKI, higher levels of Glucose, Lactate, and Triglyceride, and a greater utilization of continuous renal replacement therapy (all P<0.05). Additionally, the TyG index levels were significantly higher in the non-survivor group compared to the survivor group (10.28±1.10 vs. 9.71±0.95, P<0.001). S1 Fig illustrates the distribution of the TyG index stratified by the mortality status of all-cause in-hospital mortality (Additional File 3, S1a Fig) and ICU mortality(Additional File 4, S1b Fig).

## Primary outcomes

The incidence of primary outcomes among groups was analyzed using Kaplan-Meier survival analysis curves, based on the TyG index quartiles as presented in Fig 2.

It was observed that patients with a higher TyG index had a higher risk of hospital(log-rank P<0.0001) and ICU death (log-rank P<0.0001). We conducted an assessment of the clinical efficacy of the TyG index using ROC analysis. However, the AUC of the TyG index did not exhibit satisfactory performance in predicting hospital mortality (AUC: 0.628,95%CI:0.570-0.687) (Additional File 5, S2 a Fig) and ICU mortality (AUC: 0.656,95% CI:0.586-0.727) (Additional File 6, S2 b Fig).

S3 Table (Additional File 7, S3 Table) presents the results of the binary logistic regression conducted to assess the risk of all-cause death in critically ill patients with acute pancreatitis. The variables that showed significance in the univariate analysis (p<0.05) and factors suggested by clinicians and based on clinical experience were included as independent variables in the binary logistic regression analysis. The influential factors identified in the analysis were age, TyG Index, Anion gap, and APSIII. The Cox hazard regression model was used to account for the influence of covariates on the outcome. Three models were employed: Model 1, which was unadjusted; Model 2, which was adjusted for age, gender, and BMI; and Model 3, which was adjusted for Age, BMI, Gender, WBC, PLT, HGB, Glu, Lac, Cre, K, TG, SOFA, AKI_48h, Hypertension, Heart failure, sepsis, Diabete, and Respiratory failure. Cox proportional risk analysis was conducted to investigate the association between the TyG index and hospital mortality. The results revealed a significant association between patients in the higher quartile of TyG index and an increased risk of hospital death in all three established Cox proportional hazards models: unadjusted HR(95%CI) of 1.23（1.04-1.46), P=0.016, partly adjusted HR(95%CI) of 1.21（1.02-1.44, P=0.031, and fully adjusted HR(95%CI) of 1.38（1.03-1.87, P=0.033), these

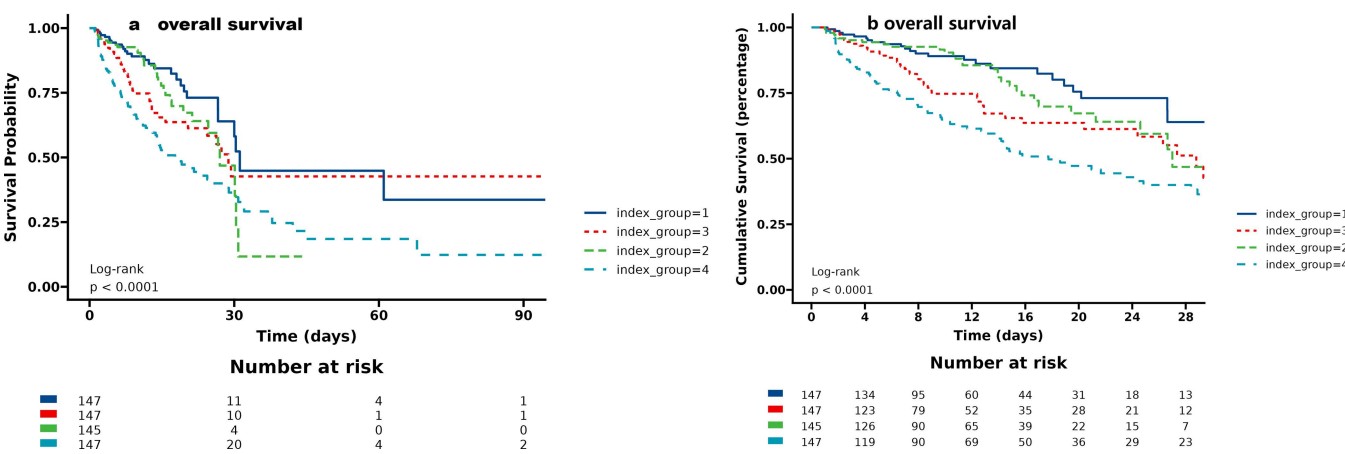

**Fig 2. Kaplan–Meier survival analysis curves for all-cause mortality. Footnote TyG index quartiles: Q1(8.56-9.20), Q2(9.20-9.95), Q3(9.95-10.57), Q4(10.57-11.09). Kaplan–Meier curves showing cumulative probability of all-cause mortality according to groups in hostital (a) and ICU (b).**

results were observed when the TyG index was considered as a continuous variable. When TyG index was treated as a nominal variable, the unadjusted model showed a HR(95%CI) of 1.92 (1.11 - 3.31), P=0.02, the partly adjusted model showed a HR(95%CI) of 1.85 (1.06 - 3.21), P=0.029, and the fully adjusted model showed a HR(95%CI) of 1.93 (1.03 - 3.60), P=0.039 compared to subjects in the lowest quartile. These findings indicate a trend of increasing risk with higher TyG index (Table 3; Fig 3a). Similar results were observed in the multivariate Cox proportional risk analysis of the TyG index and ICU mortality (Table 3; Fig 3b).

The study employed a restricted cubic splines regression model to investigate the relationship between TyG index and the risk of hospital mortality and ICU mortality. The findings

**Table 3. Cox proportional hazard ratios (HR) for all-cause mortality.**

| Variable | Model 1 | | | Model 2 | | | Model 3 | | |
|---|---|---|---|---|---|---|---|---|---|
| | HR(95%C) | P | P for trend | HR (95%CI) | P | P for trend | HR(95%C) | P | P for trend |
| Hospital mortality | | | 0.009 | | | 0.016 | | | 0.023 |
| TyG as continuous | 1.2 (1.0-1.4) | 0.016 | | 1.2 (1.2-1.4) | 0.031 | | 1.3 (1.0-1.8) | 0.033 | |
| Q1 (n = 147) | Ref | | | Ref | | | Ref | | |
| Q2 (n = 145) | 1.3 (0.7-2.4) | 0.404 | | 1.2 (0.6- 2.3) | 0.461 | | 1.1 (0.6-2.2) | 0.636 | |
| Q3 (n = 147) | 1.7 (0.9-3.1) | 0.077 | | 1.6 (0.9-2.9) | 0.105 | | 1.6 (0.9-3.1) | 0.106 | |
| Q4 (n = 147) | 1.9 (1.1-3.3) | 0.020 | | 1.8 (1.0- 3.2) | 0.029 | | 1.9 (1.0-3.6) | 0.039 | |
| ICU mortality | | | 0.002 | | | 0.005 | | | 0.017 |
| TyG as continuous | 1.3 (1.0-1.6) | 0.007 | | 1.3 (1.0-1.6) | 0.014 | | 1.6 (1.1-2.4) | 0.012 | |
| Q1 (n = 147) | Ref | | | Ref | | | Ref | | |
| Q2 (n = 145) | 1.3 (0.6-3.2) | 0.453 | | 1.3 (0.5 - 3.1) | 0.470 | | 1.6 (0.7-3.9) | 0.244 | |
| Q3 (n = 147) | 1.5 (0.7-3.4) | 0.270 | | 1.4 (0.6- 3.1) | 0.399 | | 1.9 (0.8-4.5) | 0.111 | |
| Q4 (n = 147) | 2.8 (1.3-5.6) | 0.004 | | 2.7 (1.3-5.53) | 0.007 | | 3.4 (1.5-7.7) | 0.002 | |

Modle 1: unadjusted.

Model 2: adjusted for Age, BMI, Gender.

Model 3: adjusted for Age, BMI, Gender, WBC, PLT, HGB, Glu, Lac, Cre, K, TG, SOFA, AKI_48h, Hypertension, Heart failure, sepsis, Diabete, and Respiratory failure.

TyG index: Q1(8.56-9.20), Q2(9.20-9.95), Q3(9.95-10.57), Q4(10.57-11.09).

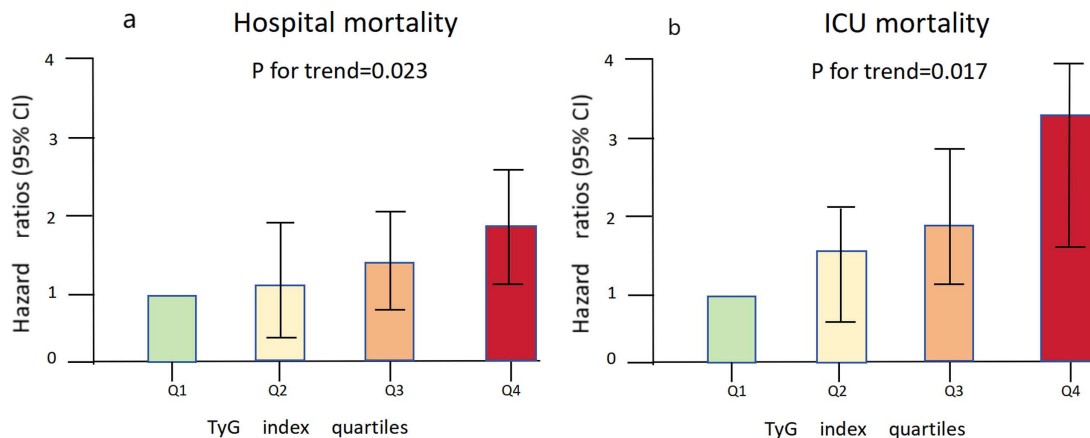

**Fig 3. (a), (b): Hazard ratios (95% CIs) for hospital mortality according to TyG index quartiles after adjusting for age, gender, BMI, WBC, PLT, HGB, FBG, Lac, Cre, K, TG, SOFA, AKI_48h, Hypertension, Heart failure, sepsis, Diabete, and Respiratory failureError bars indicate 95% CIs. The frst quartile is the reference.**

indicated that both hospital mortality and ICU mortality increased linearly as the TyG index increased (P for non-linearity=0.572 and P for non-linearity=0.186, respectively) (Fig 4).

### Subgroup analysis.

To investigate the relationship between TyG index and in-hospital mortality and ICU mortality, stratified analyses were performed based on various factors including age, gender, BMI, AKI_48h, respiratory failure, sepsis and diabete. The risk of hospital mortality in different subgroups of acute pancreatitis patients was analyzed. The results showed that those aged >65 years [HR (95% CI) 1.83(1.27-2.77)] had a higher risk of hospital mortality. Those with respiratory failure [HR (95% CI) 0.44(0.29-0.64)], and those with sepsis [HR (95% CI) 0.63 (0.37-0.89)] had a lower risk of hospital mortality (Fig 5a).

Interestingly, when analyzing ICU mortality, the TyG index showed only those with respiratory failure [HR (95% CI) 0.58(0.35-0.96)] had a lower risk of ICU. The association between TyG index and ICU mortality did not differ among other subgroups (Fig 5b).

## Discussion

To the best of our knowledge, this study represents the first investigation into the association between the TyG index and all-cause mortality in critically ill patients with acute pancreatitis. The findings from this study suggest that a higher TyG index is significantly correlated with increased ICU and hospital mortality rates among critically ill patients with AP. Notably, this association remained statistically significant even after adjusting for potential confounding risk factors. Consequently, these results indicate that the TyG index holds promise as a valuable decision-making tool for clinicians and may serve as an independent prognostic factor in critically ill patients with AP.

Metabolic syndrome is known to increase the incidence of acute AP and the risk of severe acute pancreatitis (SAP). Triglycerides (TG) and glucose (GLU) are commonly used as markers for metabolic diseases. It is evident that hypertriglyceridemia plays a critical role in AP. Nagy et al. conducted a study on the clinical process of AP and identified high blood glugose as a metabolic factor that contributes to toxic side effects, ultimately impacting the severity

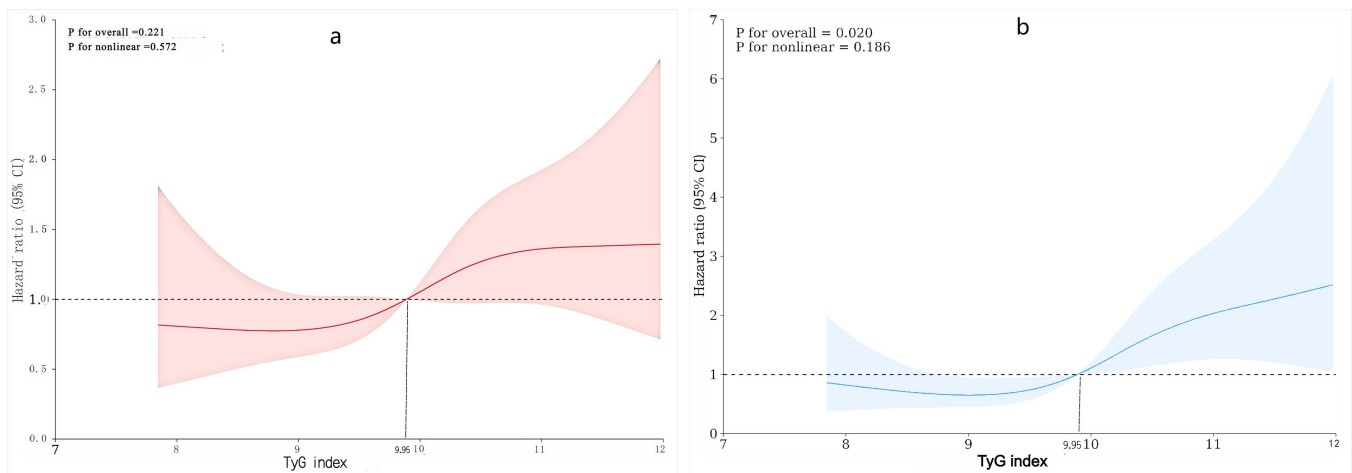

**Fig 4. (a), (b): Restricted cubic spline curve for the TyG index hazard ratio.** Heavy central lines represent the estimated adjusted hazard ratios, with shaded ribbons denoting 95% confdence intervals. TyG index 9.95 was selected as the reference level represented by the vertical dotted lines. The horizontal dotted lines represent the hazard ratio of 1.0.

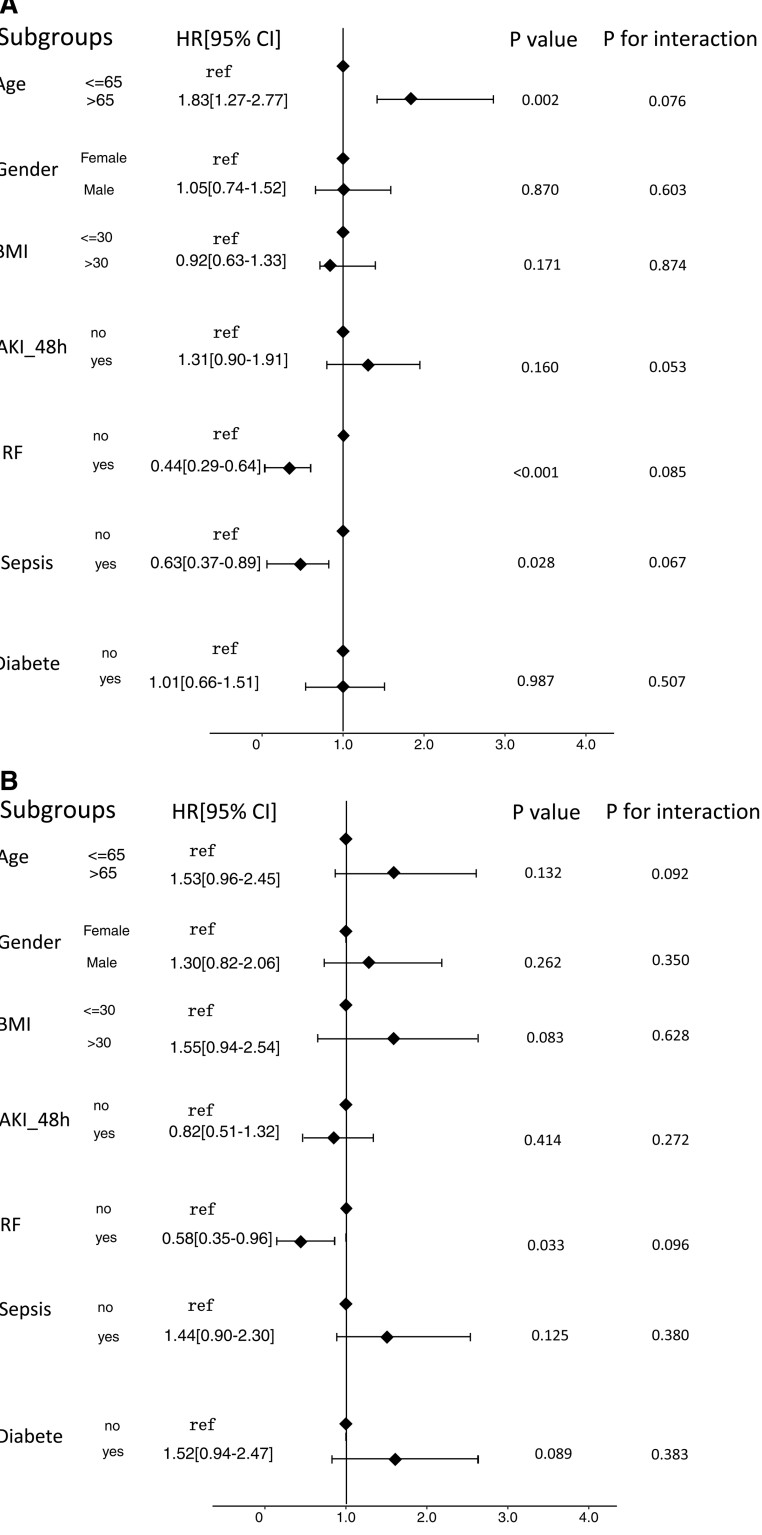

**Fig 5.** **A** Forest plots of hazard ratios for the hospital mortality in different subgroups. HR, hazard ratio; CI, confidence interval; BMI, body mass index; AKI_48hr: Acute Kidney Injury within 48 h; RF: Respiratory failure. **B** Forest plots of hazard ratios for the ICU mortality in different subgroups. HR, hazard ratio; CI, confidence interval; BMI, body mass index; AKI_48hr: Acute Kidney Injury within 48 h; RF: Respiratory failure.

and mortality of AP [20]. The TyG index, which includes TG and FBG(fast blood glucose), has been proposed as a potential marker for metabolic disorders, atherosclerotic disease, cardiovascular disease, and sepsis [10,12,21,22].

Several clinical studies have examined how an elevated TyG index can lead to higher mortality rates in critically ill patients and those with infectious diseases in various populations. Lee et al. discovered that the TyG index might serve as a useful tool in predicting immediate functional outcomes for critically ill stroke patients undergoing reperfusion therapy [23]. Yang et al. conducted a study and discovered that the TyG index was identified as an independent risk factor for mortality in patients who experienced cardiac arrest during their hospitalization [24]. Furthermore, the TyG index has demonstrated potential in predicting adverse cardiovascular events among individuals diagnosed with coronary artery disease [25,26]. In the general population, a higher TyG index has been associated with an increased prevalence of respiratory symptoms, heightened susceptibility to chronic lung disease, and deterioration of lung function. [27]. In previous research, it has been consistently observed that there is a correlation between an elevated TyG index and a higher risk of in-hospital mortality in critically ill patients. Specifically, it has been found that for each unit increase in the TyG index, the risk of in-hospital mortality increases by approximately 30% or more [28,29]. Building upon these findings, our study focused on AP patients and found that the risk of in-hospital mortality increased by 38% for these patients with each unit increase in the TyG index. These results align with previous research, further supporting the notion that an elevated TyG index is associated with higher mortality rates in critically ill patients.

The exact biological mechanism underlying the relationship between the TyG index and the development and progression of cerebrovascular disease and mortality is still uncertain. One possible pathway could be related to insulin resistance(IR). A previous study has shown that glucose may indicate IR from the liver, while TG reflects IR from adipose tissue [30]. (Comment 2)A correlation exists between the TyG index and insulin resistance, as insulin resistance weakens the body's response to insulin, impacting the liver's ability to process glucose and fat metabolism [31,32]. In cases of insulin resistance, the liver may increase triglyceride production while decreasing glucose utilization, leading to elevated levels of triglycerides and blood glucose in the bloodstream [33]. Oxidative stress triggered by high blood glucose levels can exacerbate the inflammatory response in acute pancreatitis (AP) and serve as a standalone prognostic indicator for AP [20]. Additionally, elevated triglyceride levels can worsen pancreatic ischemia, hypoxia, microcirculation abnormalities, and systemic inflammatory response, all of which can complicate AP [34]. Therefore, an elevated TyG index can serve as an indicator of insulin resistance. The TyG index is highly sensitive and specific in evaluating IR [35]. Firstly, IR has been widely demonstrated to be closely related to inflammatory response, immune dysregulation, endothelial dysfunction, aoxidative stress, and coagulation imbalance [36–38]. Based on the baseline data, we have observed significant differences in SOFA scores among patients in different TyG index groups. This suggests a strong association between the TyG index and disease severity. An elevated TyG index is associated with cardiovascular diseases, and the presence of cardiovascular diseases in AP is a risk factor contributing to adverse patient outcomes. However, in this study, no differences were observed in the presence of cardiovascular diseases among the various TyG index level groups. Furthermore, the TyG index may reflect the metabolism of glycosylation products and platelet reactivity, leading to endothelium-dependent vasodilation [39].

In our current investigation, an important linear correlation was identified between the TyG index and mortality during hospitalization. This indicates that the TyG index has the potential to be a valuable tool in identifying individuals with a high risk of mortality. Therefore, maintaining optimal levels of triglycerides (TG) and glucose is crucial in order to reduce the occurrence

of major adverse clinical outcomes in the future. Proper management of the TyG index is also important for this purpose. Furthermore, our subgroup analysis revealed that the TyG index has consistent predictive value for in-hospital mortality and ICU mortality in both male and female patients. Nevertheless, no correlation was found between the TyG index and in-hospital mortality for the patients with diabetes and AKI included in this study. This finding could potentially be explained by reverse causation, which suggests that individuals who have previously been diagnosed with these comorbidities may have a higher likelihood of receiving appropriate medical care or engaging in healthy lifestyle practices. Additionally, our research revealed that the relationship between the TyG index and in-hospital mortality seemed to be more pronounced among patients aged 65 years or older. Interestingly, no statistically significant findings were observed for ICU mortality when stratifying by age. Nonetheless, it is worth emphasizing that the reduced sample size resulting from stratified analysis might have led to a decrease in the magnitude of the effect. This decrease in effect size could be a contributing factor to the absence of significant results. (The study's limitations)However, our study also had several limitations. Firstly, as this study was retrospective in nature, it was unable to definitively establish causality. Despite the use of multivariate adjustment and subgroup analyses, there is still a possibility of residual confounding factors influencing the clinical outcomes. (Comment 3)Secondly, it is important to note that the TyG index was not continuously monitored throughout the study, our investigation solely focused on evaluating the prognostic value of the baseline TyG index for critically ill patients with acute pancreatitis, disregarding any dynamic changes in the TyG index. Thirdly, the progression of acute pancreatitis can potentially impact lipid metabolism and lead to fluctuations in blood glucose levels. The TyG index obtained from the initial measurements of glucose and triglyceride might not completely reflect the overall insulin resistance in the body. It is important to consider this limitation when interpreting the results.(Comment 5) Fourthly, due to the limitations of the MIMIC-IV database, we were unable to obtain the specific times at which fasting blood glucose and triglycerides were measured. Blood glucose and triglyceride measurements were not standardized for each patient, different therapy and food intake may also affect the frequency of blood glucose and triglyceride measurements. Additionally, certain confounding factors such as inflammatory markers, metabolic syndrome parameters, nutritional state parameters, and Acute Physiology and Chronic Health Evaluation II (APACHE II) were not thoroughly taken into account, which could potentially influence the findings. (Comment 1)Finally, this study had a sample size of moderate magnitude, a relatively short follow-up duration, and was solely conducted at a single center, potentially introducing selection bias. Therefore, more prospective studies with a larger cohort of subjects are needed to support the present findings.

## Conclusions

In this study, we expanded the application of the TyG index to critically ill patients with AP and found that it can serve as a useful tool for risk stratification of in-hospital and ICU mortality in these patients. (Comment 4) However, the study sample is specific to critically ill patients with acute pancreatitis, and the results may differ in other contexts or populations. Our findings indicate that the TyG index can be useful in identifying insulin resistance at an early stage in critically ill patients with AP, thereby improving risk assessment and guiding subsequent interventions. Whether better control of TyG index can improve clinical prognosis requires further study.

## Supporting information

**S1 Table. Missing number for risk variables and outcome variables.**
(DOCX)

**Table S2.  Characteristics and outcomes of participants categorized by TyG index.**
(DOCX)

**S3Table.  Binary logistic regression analysis of the factors influencing all-cause mortality of the study population.**
(DOCX)

**S1 a Fig.  The distribution of the TyG index stratified by the mortality status of all-cause in-hospital mortality.**
(PNG)

**S1 b Fig.  The distribution of the TyG index stratified by the mortality status of all-cause in-ICU mortality.**
(PNG)

**S2 a Fig.  ROC of the TyG index in predicting hospital mortality.**
(PNG)

**S2 b Fig.  ROC of the TyG index in predicting ICU mortality.**
(PNG)

## Author contributions

**Conceptualization:** Jian Liao.

**Data curation:** Dingyu Lu.

**Formal analysis:** Jian Liao.

**Investigation:** Dingyu Lu.

**Methodology:** Jian Liao.

**Resources:** Dingyu Lu.

**Software:** Dingyu Lu.

**Supervision:** Maojuan Wang.

**Validation:** Hong Xie.

**Writing – original draft:** Jian Liao.

**Writing – review & editing:** Maojuan Wang.

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
