## [Decision Letter · Decision Letter 0]

20 May 2024

PONE-D-24-05736The role of TyG index as a predictor of all-cause mortality in hospitalized patients with acute pancreatitis:a retrospective study utilizing the MIMIC-IV database.PLOS ONE

Dear Dr. Liao,

Thank you for submitting your manuscript to PLOS ONE. After careful consideration, we feel that it has merit but does not fully meet PLOS ONE’s publication criteria as it currently stands. Therefore, we invite you to submit a revised version of the manuscript that addresses the points raised during the review process.Please, answer all questions presented by the reviewers and  submit your revised manuscript by Jul 04 2024 11:59PM. If you will need more time than this to complete your revisions, please reply to this message or contact the journal office at plosone@plos.org . Please include the following items when submitting your revised manuscript:

We look forward to receiving your revised manuscript.

Kind regards,

Fabio Vasconcellos Comim, MD,PhD

Academic Editor

PLOS ONE

Journal Requirements:

2. PLOS requires an ORCID iD for the corresponding author in Editorial Manager on papers submitted after December 6th, 2016. Please ensure that you have an ORCID iD and that it is validated in Editorial Manager. To do this, go to ‘Update my Information’ (in the upper left-hand corner of the main menu), and click on the Fetch/Validate link next to the ORCID field. This will take you to the ORCID site and allow you to create a new iD or authenticate a pre-existing iD in Editorial Manager. Please see the following video for instructions on linking an ORCID iD to your Editorial Manager account: https://www.youtube.com/watch?v=_xcclfuvtxQ".

4. Please remove your figures from within your manuscript file, leaving only the individual TIFF/EPS image files, uploaded separately. These will be automatically included in the reviewers’ PDF.

5. Please include your tables as part of your main manuscript and remove the individual files. Please note that supplementary tables (should remain/ be uploaded) as separate ""supporting information"" files.

Additional Editor Comments:

Make very clear the study's limitations in the discussion

Reviewers' comments:

Reviewer's Responses to Questions

**Comments to the Author**

1. Is the manuscript technically sound, and do the data support the conclusions?

Reviewer #1: Partly

2. Has the statistical analysis been performed appropriately and rigorously? 

Reviewer #1: Yes

3. Have the authors made all data underlying the findings in their manuscript fully available?

Reviewer #1: Yes

4. Is the manuscript presented in an intelligible fashion and written in standard English?

Reviewer #1: Yes

5. Review Comments to the Author

Reviewer #1: Dear authors and editorial board,

I congratulate you all on your work and the hypotheses presented, as well as on the good adequacy of the English language. Your pursuit of further evidence is also commendable. However, it is important to elucidate some significant biases in this retrospective study. I suggest evaluating the following aspects:

1. **Selection Bias**: Given that this study is retrospective, I suggest elucidating the selection bias of patients due to the retrospective nature of the study.

2. **Accuracy of the TyG Index**: The TyG index was calculated from initial measurements of glucose and triglycerides. I suggest clarifying how these measurements accurately reflect insulin resistance or the overall metabolic state throughout the patient's hospitalization. Additionally, it is important to consider that acute states of pancreatitis can falsely lower triglyceride levels, which can be a significant bias. The inflammatory state of pancreatitis can correspond to a Systemic Inflammatory Response Syndrome (SIRS), which can elevate other markers and, by itself, increase glucose levels and temporarily reduce insulin resistance.

3. **Lack of Continuous Monitoring**: Evaluating the TyG index at a single point in time may represent a more severe clinical situation for one patient compared to another. Therefore, I suggest that the analysis be dynamic to assess the real effect on the patient's metabolic state and its impact on mortality.

4. **Generalization of Findings**: The findings may not be generalizable to all populations. The study sample is specific to critically ill patients with acute pancreatitis, and the results may differ in other contexts or populations. Additionally, the etiology of pancreatitis needs to be better established.

5. **Variability in Laboratory Measurements**: Variability in laboratory measurements, fasting times, whether the patient has a patent oral route, and other methods such as the timing of sample collection can affect glucose and triglyceride levels, impacting the accuracy of the TyG index. This needs to be better established.

6. **Disease Severity Factors**: Factors such as disease severity, measured by the APACHE II score (Acute Physiology and Chronic Health Evaluation II), were not fully considered. These factors can significantly impact the study's findings, representing an important bias in this work.

Thank you for your attention and for the opportunity to contribute to the improvement of this study.

Sincerely,

6. PLOS authors have the option to publish the peer review history of their article (what does this mean? ). If published, this will include your full peer review and any attached files.

**Do you want your identity to be public for this peer review?** For information about this choice, including consent withdrawal, please see our Privacy Policy .

Reviewer #1: **Yes:**

---

## [Author Response · Author response to Decision Letter 0]

22 May 2024

Dear Editor and reviewers,

Thank you for reviewing our manuscript and for the constructive comments, which greatly helped us to improve the manuscript (PONE-D-24-05736). We appreciate the promoting comments to our study, and we have accepted and revised as recommended in this revised manuscript. We highlighted all the revisions in yellow/red colour.

We would like to express our great appreciation to you and reviewers for comments on our paper. Looking forward to hearing from you.

Thank you and best regards. 

The main corrections in the paper and the responds to the reviewer’s comments are as following:

Reviewer #1:

Comment 1:**Selection Bias**: Given that this study is retrospective, I suggest elucidating the selection bias of patients due to the retrospective nature of the study.

Response: It is really true as Reviewer suggested that the selection bias of patients should be explained.There is a potential selection bias to this retrospective study. Therefore, more prospective studies with a larger cohort of subjects are needed to support the present findings. We have mentioned this in the last paragraph of Discussion marked in yellow.f

Comment 2:**Accuracy of the TyG Index**: The TyG index was calculated from initial measurements of glucose and triglycerides. I suggest clarifying how these measurements accurately reflect insulin resistance or the overall metabolic state throughout the patient's hospitalization. Additionally, it is important to consider that acute states of pancreatitis can falsely lower triglyceride levels, which can be a significant bias. The inflammatory state of pancreatitis can correspond to a Systemic Inflammatory Response Syndrome (SIRS), which can elevate other markers and, by itself, increase glucose levels and temporarily reduce insulin resistance.

Response: We have reviewed the relevant literature again and found a correlation exists between the TyG index and insulin resistance, as insulin resistance weakens the body's response to insulin, impacting the liver's ability to process glucose and fat metabolism. In cases of insulin resistance, the liver may increase triglyceride production while decreasing glucose utilization, leading to elevated levels of triglycerides and blood glucose in the bloodstream. Oxidative stress triggered by high blood glucose levels can exacerbate the inflammatory response in acute pancreatitis (AP) and serve as a standalone prognostic indicator for AP. Additionally, elevated triglyceride levels can worsen pancreatic ischemia, hypoxia, microcirculation abnormalities, and systemic inflammatory response, all of which can complicate AP. Therefore, an elevated TyG index can serve as an indicator of insulin resistance. We have mentioned this in the third paragraph of Discussion marked in yellow and reference [31,32,33,34] has been added.

Comment 3: *Lack of Continuous Monitoring**: Evaluating the TyG index at a single point in time may represent a more severe clinical situation for one patient compared to another. Therefore, I suggest that the analysis be dynamic to assess the real effect on the patient's metabolic state and its impact on mortality.

Response: Due to the limitations of the MIMIC-IV database itself, we were unable to obtain dynamically changing fasting blood glucose and triglyceride levels. We tried to collect these two indicators in other time periods, but the data loss exceeded 50%, making it impossible to conduct rigorous statistical analysis. We hope the reviewer understands these difficulties.We have mentioned this in the last paragraph of Discussion marked in yellow.

Comment 4: **Generalization of Findings**: The findings may not be generalizable to all populations. The study sample is specific to critically ill patients with acute pancreatitis, and the results may differ in other contexts or populations. Additionally, the etiology of pancreatitis needs to be better established.

Response: It is really true as Reviewer pointed out that the findings may not be generalizable to all populations, We have explained this shortcoming in the Conclusion marked in yellow. The MIMIC-IV database only includes patients with biliary pancreatitis and alcoholic pancreatitis, not common hyperlipidemia pancreatitis. This limits the accuracy of etiological data collection. Making hasty generalizations could result in the loss of valuable clinical data and authenticity. Our center is currently conducting a prospective study to gather data on acute pancreatitis cases from various causes. We anticipate that latter data analysis will provide statistically significant results.

Comment 5: **Variability in Laboratory Measurements**: Variability in laboratory measurements, fasting times, whether the patient has a patent oral route, and other methods such as the timing of sample collection can affect glucose and triglyceride levels, impacting the accuracy of the TyG index. This needs to be better established.

Response: It is really true as Reviewer pointed out that variability in laboratory measurements, fasting times, the timing of sample collection can affect glucose and triglyceride levels, impacting the accuracy of the TyG index. We also noticed this flaw. Due to the limitations of the MIMIC-IV database,we were unable to obtain the specific times at which fasting blood glucose and triglycerides were measured.Blood glucose and triglyceride measurements were not standardized for each patient, different therapy and food intake may also affect the frequency of blood glucose and triglyceride measurements. We have mentioned this in the last paragraph of Discussion marked in yellow.

Comment 6: **Disease Severity Factors**: Factors such as disease severity, measured by the APACHE II score (Acute Physiology and Chronic Health Evaluation II), were not fully considered. These factors can significantly impact the study's findings, representing an important bias in this work.

Response: Some confounding factors, including metabolic syndrome parameters,Acute Physiology and Chronic Health Evaluation II (APACHE II), nutritional state parameters, and infammatory markers, were not thoroughly considered. It is a pity that there is no score for APACHE II recorded in the MIMIC-IV database.This may have an impact on the results. Our center is currently conducting a prospective study to gather data on score for APACHE II. We anticipate that latter data analysis will provide statistically significant results.

Dear Editor and reviewers,We hope that the revision is acceptable, and your favorable consideration of our manuscript is greatly appreciated. Best regards.

---

## [Decision Letter · Decision Letter 1]

5 Aug 2024

The role of TyG index as a predictor of all-cause mortality in hospitalized patients with acute pancreatitis:a retrospective study utilizing the MIMIC-IV database.

PONE-D-24-05736R1

Dear Dr. Liao,

We’re pleased to inform you that your manuscript has been judged scientifically suitable for publication and will be formally accepted for publication once it meets all outstanding technical requirements.

Kind regards,

Fabio Vasconcellos Comim, MD,PhD

Academic Editor

PLOS ONE

**Comments to the Author**

1. If the authors have adequately addressed your comments raised in a previous round of review and you feel that this manuscript is now acceptable for publication, you may indicate that here to bypass the “Comments to the Author” section, enter your conflict of interest statement in the “Confidential to Editor” section, and submit your "Accept" recommendation.

Reviewer #2: All comments have been addressed

2. Is the manuscript technically sound, and do the data support the conclusions?

Reviewer #2: Yes

3. Has the statistical analysis been performed appropriately and rigorously? 

Reviewer #2: Yes

4. Have the authors made all data underlying the findings in their manuscript fully available?

Reviewer #2: No

5. Is the manuscript presented in an intelligible fashion and written in standard English?

Reviewer #2: Yes

6. Review Comments to the Author

Reviewer #2: I have no additional comment.

xxxxxxxxxxxxxxxxxxxxxxxxxxxxxxxxxxxxxxxxxxxxxxxxxxxxxxxxxxxxxxxxxxxxxxxxxxxxxxxxxxxxxxxxxx

7. PLOS authors have the option to publish the peer review history of their article (what does this mean? ). If published, this will include your full peer review and any attached files.

**Do you want your identity to be public for this peer review?** For information about this choice, including consent withdrawal, please see our Privacy Policy .

Reviewer #2: No

---

## [Editor Report · Acceptance letter]

PONE-D-24-05736R1

PLOS ONE

Dear Dr. Liao,

I'm pleased to inform you that your manuscript has been deemed suitable for publication in PLOS ONE. Congratulations! Your manuscript is now being handed over to our production team.

Kind regards,

on behalf of

Prof Fabio Vasconcellos Comim

Academic Editor

PLOS ONE